# The Evolution in Anxiety and Depression with the Progression of the Pandemic in Adult Populations from Eight Countries and Four Continents

**DOI:** 10.3390/ijerph18094845

**Published:** 2021-05-01

**Authors:** Mélissa Généreux, Philip J. Schluter, Elsa Landaverde, Kevin KC Hung, Chi Shing Wong, Catherine Pui Yin Mok, Gabriel Blouin-Genest, Tracey O’Sullivan, Marc D. David, Marie-Eve Carignan, Olivier Champagne-Poirier, Nathalie Pignard-Cheynel, Sébastien Salerno, Grégoire Lits, Leen d’Haenens, David De Coninck, Koenraad Matthys, Eric Champagne, Nathalie Burlone, Zeeshan Qadar, Teodoro Herbosa, Gleisse Ribeiro-Alves, Ronald Law, Virginia Murray, Emily Ying Yang Chan, Mathieu Roy

**Affiliations:** 1Department of Community Health Sciences, Faculté de Médecine et des Sciences de la Santé, Université de Sherbrooke, Sherbrooke, QC J1H 5N4, Canada; elsa.landaverde@umontreal.ca; 2School of Health Sciences, University of Canterbury-Te Whare Wananga o Waitaha, Christchurch 8140, New Zealand; philip.schluter@canterbury.ac.nz; 3Collaborating Centre for Oxford University and CUHK for Disaster and Medical Humanitarian Response, JC School of Public Health and Primary Care, Chinese University of Hong Kong, Ngan Shing Street 30-32, Hong Kong, China; kevin.hung@cuhk.edu.hk (K.K.H.); cswong@cuhk.edu.hk (C.S.W.); catherine.mok@cuhk.edu.hk (C.P.Y.M.); emily.chan@cuhk.edu.hk (E.Y.Y.C.); 4School of Applied Politics, Faculté des Lettres et Sciences Humaines, Université de Sherbrooke, Sherbrooke, QC J1K 2R1, Canada; gabriel.blouin-genest@usherbrooke.ca; 5Interdisciplinary School of Health Sciences, Faculty of Health Sciences, University of Ottawa, Ottawa, ON K1N 7K4, Canada; tosulliv@uottawa.ca; 6Department of Communication, Faculté de Lettres et Sciences Humaines, Université de Sherbrooke, Sherbrooke, QC J1K 2R1, Canada; marc.d.david@usherbrooke.ca (M.D.D.); marie-eve.carignan@usherbrooke.ca (M.-E.C.); olivier.champagne-poirier@usherbrooke.ca (O.C.-P.); 7Académie du Journalisme et des Médias, Université de Neuchâtel, 2000 Neuchâtel, Switzerland; nathalie.pignard-cheynel@unine.ch; 8Medi@Lab, Université de Genève, Boulevard du Pont-d’Arve 40, 1205 Genève, Switzerland; Sebastien.Salerno@unige.ch; 9Institut Langage et Communication, Université catholique de Louvain, 1348 Louvain-la-Neuve, Belgium; gregoire.lits@uclouvain.be; 10Institute for Media Studies, KU Leuven, 3000 Leuven, Belgium; leen.dhaenens@kuleuven.be; 11Centre for Sociological Research, KU Leuven, 3000 Leuven, Belgium; david.deconinck@kuleuven.be (D.D.C.); koen.matthijs@kuleuven.be (K.M.); 12School of Political Studies, Faculty of Social Sciences, University of Ottawa, Ottawa, ON K1N 6N5, Canada; echampagne@uottawa.ca (E.C.); nburlone@uottawa.ca (N.B.); 13National Collaborating Centre for Infectious Diseases, Rady Faculty of Health Sciences, University of Manitoba, Winnipeg, MB R3E 0T5, Canada; sheikh.qadar@umanitoba.ca; 14Department of Emergency Medicine, College of Medicine, University of Philippines, Manille Grand, Manila 1000, Philippines; ted.herbosa@gmail.com; 15Centro Universitário de Brasília, Brasilia 70850-090, Brazil; gleisse@yahoo.com; 16Department of Health, Manila, Manila 2932, Philippines; ronlawmd@gmail.com; 17Public Health England, London SE1 8UG, UK; virginia.murray@phe.gov.uk; 18Department of Family Medicine & Emergency Medicine, Faculté de Médecine et des Sciences de la Santé, Université de Sherbrooke, Sherbrooke, QC J1H 5N4, Canada; Mathieu.roy7@usherbrooke.ca

**Keywords:** pandemic, psychosocial impacts, sense of coherence

## Abstract

Nearly a year after the classification of the COVID-19 outbreak as a global pandemic, it is clear that different factors have contributed to an increase in psychological disorders, including public health measures that infringe on personal freedoms, growing financial losses, and conflicting messages. This study examined the evolution of psychosocial impacts with the progression of the pandemic in adult populations from different countries and continents, and identified, among a wide range of individual and country-level factors, which ones are contributing to this evolving psychological response. An online survey was conducted in May/June 2020 and in November 2020, among a sample of 17,833 adults (Phase 1: 8806; Phase 2: 9027) from eight countries/regions (Canada, the United States, England, Switzerland, Belgium, Hong Kong, the Philippines, New Zealand). Probable generalized anxiety disorder (GAD) and major depressive episode (MDE) were assessed. The independent role of potential factors was examined using multilevel logistic regression. Probable GAD or MDE was indicated by 30.1% and 32.5% of the respondents during phases 1 and 2, respectively (a 7.9% increase over time), with an important variation according to countries/regions (range from 22.3% in Switzerland to 38.8% in the Philippines). This proportion exceeded 50% among young adults (18–24 years old) in all countries except for Switzerland. Beyond young age, several factors negatively influenced mental health in times of pandemic; important factors were found, including weak sense of coherence (adjusted odds ratio aOR = 3.89), false beliefs (aOR = 2.33), and self-isolation/quarantine (aOR = 2.01). The world has entered a new era dominated by psychological suffering and rising demand for mental health interventions, along a continuum from health promotion to specialized healthcare. More than ever, we need to innovate and build interventions aimed at strengthening key protective factors, such as sense of coherence, in the fight against the adversity caused by the concurrent pandemic and infodemic.

## 1. Introduction

Since its first identification as a cluster of atypical pneumonias in Wuhan, China in December 2019 [1], the COVID-19 virus has spread to 224 countries around the globe [2]. Nearly a year after the classification of the outbreak as a global pandemic by the World Health Organization on 11th March 2020 [1], the disease counts more than 119 million reported cases and over 2 million deaths globally (14 March 2021) [2]. Protecting people from infection, imposition of public health measures that infringe on personal and collective freedoms, growing financial losses, and conflicting messages from authorities are among the major stressors having contributed to the widespread emotional distress and increased risk of psychological disorders associated with COVID-19. With growing knowledge of the initial impacts of the pandemic on psychological health and well-being and the increasing importance of the infodemic, an information overload observed during large-scale epidemics [3], research now shifts towards the longitudinal surveillance of these adverse effects and investigating mitigation strategies [4].

Although signs of hope are appearing among the population, due to rapid advancements in the development, production and distribution of the COVID-19 vaccine, the psychological repercussions of COVID-19 continue to deepen as many countries face the second wave, and even a third one in some cases, of the pandemic. Extensive research on disaster mental health has established that emotional distress is ubiquitous in affected local communities; a finding certain to be echoed during the pandemic, but this time among virtually the entire world population, as no one is immune to its collateral damages [5,6]. Throughout the world researchers have found consistent evidence that the pandemic has triggered a surge in psychopathological disorders and symptoms. In Canada, a study exploring the changes in self-reported mental health noted a deterioration in mental health in 38.2% of respondents during the first wave of the pandemic. Individuals with pre-existing mental health conditions, with disabilities and with annual household incomes <25,000$ CAD were more likely to be affected [7]. In the United States, about 29% of the adult population reported some depression/anxiety symptoms, with symptoms deteriorating over the month of March [8]. The key driving forces of this psychological crisis have been economic concerns, health implications and social distancing measures [8]. Researchers in Italy and Belgium found that lockdown delayed sleep timing, increased time spent in bed, and impaired sleep quality, especially for those who perceived the pandemic as highly stressful [9]. A study in China reported COVID-19-related increases in generalized anxiety, which were more pronounced among younger people (<25 years) compared to older age groups [10]. In another study, this time conducted in New Zealand, suicidal ideation was reported by 6% of participants during the COVID-19 lockdown, with 2% reporting making plans for suicide, 2% reporting suicide attempts and with suicidality levels the highest in those aged 18–34 [11]. Health care workers also seem to be particularly affected. A meta-analysis of 13 studies of mental health among healthcare workers found that almost a quarter exhibited elevated COVID-19-related symptoms of anxiety (23.2%) and depression (22.8%) [12].

Measures implemented to control the spread of the COVID-19 virus vary greatly by countries and regions [13]. Countries across the globe faced different epidemiological situations, and even those impacted in a similar manner chose to respond to the pandemic in different ways. Early and effective containment measures have decreased infection rates [13]; however, the benefits come with huge costs in terms of negative psychological outcomes [9], especially when individuals are affected by specific stressors such as greater duration of confinement, inadequate supplies, difficulty securing medical care and financial losses [14]. The home confinement of large swaths of the population for indefinite periods, differences among the stay-at-home orders issued by various jurisdictions, and conflicting messages from government and public health authorities have most likely intensified distress. The confinement also deepened social inequalities, causing more individuals to be vulnerable to the impacts of the pandemic. Although pandemic-related factors have greatly affected the mental health of the population, their interaction with a country’s socio-cultural, historic and political context is also expected to have influenced this response.

Beyond the stressors directly related to the pandemic, it is crucial to consider the infodemic as a critical factor contributing to the adverse outcomes of the COVID-19 pandemic. Not only can mis- and disinformation negatively impact an individual’s physical and mental health, they can ultimately affect countries’ abilities to stop the pandemic, as the effectiveness of public health measures is reduced due to poor compliance [3]. As the pandemic continues to evolve, the relationship between communication strategies, media discourse and psychosocial impacts continues to strengthen. The negative influence of excessive exposure to media on mental health has been exposed in several studies. Wang et al. (2020) surveyed participants at two-time points following the commencement of the pandemic and found that increased exposure to radio reports about COVID-19 was significantly associated with higher levels of anxiety and depression [15]. A German study also suggests that increased frequency, duration and diversity of media exposure increase the risk of psychological distress, with higher levels of anxiety and depression when social media is used as the primary type of information resource [16]. Existing research also highlighted a number of COVID-19-related risks linked to information and communication failure such as confusion, misunderstanding, trust/mistrust, sense of fear, creation of conspiracy theories, denial, racial discrimination and avoidance behaviours, all potentially having harmful effects on mental health.

Various factors play important roles in coping with a highly stressful situation such as the COVID-19 pandemic and the ensuing turmoil [17]. Given that disruption, loss, and confusion due to the pandemic and the infodemic could realistically affect our everyday lives, recognizing and strengthening these protective factors might be key to mitigating the psychosocial impacts of the current crisis. These encompass not only individual psychological resources, such as one’s sense of coherence (SOC) described in the salutogenic approach of health promotion [18], but also socio-ecological factors for resilience such as family functioning, social support, social participation, and trust in healthcare institutions. These factors are known to be associated with positive mental health and well-being outcomes [19]. SOC is a psychological resource that develops over the lifetime and increases individuals’ capacity to use resistance resources to effectively deal with stressful circumstances. Individuals with a higher SOC are able to show an understanding of their stressors, are more confident in their coping abilities, and more motivated to cope with stressors, making them more resilient when faced with stressful situations [18]. This protective effect has been noted in various studies, where the buffering role of the SOC moderated the link between COVID-19 illness experiences and psychological well-being [18,20,21].

The pandemic has had, and will continue to have, profound psychological effects around the world. It is essential to understand how the adverse mental health outcomes are progressing with the perpetuation of the health crisis, and how they are influenced by the infodemic and other challenges arising in the context of the pandemic. Multilevel factors that positively or negatively contribute to the psychological response of the population must be better understood, to put in place appropriate interventions. Numerous studies have explored these factors in conjunction with prospective changes in mental health, while others demonstrated the association between COVID-19 and mental health along with their influencing factors. However, additional research should be carried out to examine factors that affect the mental health of the population over time, to inform the development of sound mitigation strategies. This study therefore aims to (1) examine the evolution in anxiety and depression with the progression of the pandemic in adult populations from different countries and continents and (2) identify, among a wide range of individual and country-level factors, which ones are positively or negatively contributing to this evolving psychological response.

## 2. Materials and Methods

### 2.1. Design

This study is the second phase of an interdisciplinary and international survey (conducted in eight countries from four continents) on the psychosocial impacts of the COVID-19 pandemic among adults and its associated risk and protective factors. This is part of a larger multidisciplinary research project funded by the Canadian Institutes of Health Research. The study was conducted in accordance with the Declaration of Helsinki, was reviewed and approved by the Research Ethics Board of the CIUSSS de l’Estrie—CHUS (HEC ref: 2020-3674). Further details concerning the goals, objectives and methods of this broad research project can be found elsewhere [17,22]. The second phase of the international survey was conducted from 6–18 November 2020, following the pilot phase conducted in Canada only in April 2020 (n = 600) and the first international phase conducted in June 2020 (n = 8806). To ensure continuity of the results, the data collection for the phase 2 was conducted among a sample of adults in the same countries or regions as the previous study: Canada, the United States of America (USA), England, Switzerland, Belgium, Hong Kong, the Philippines and New Zealand, using the same recruitment strategies.

### 2.2. Selection of Participants

To ensure continuity between the two phases of the project, the same polling firms were responsible for the recruitment of participants and coordination of the data collection. Great attention was paid to the recruitment and sampling methods, which were consistent between participating countries (or regions) and between the two phases of data collection, in order to make the data comparable. Eligible participants consisted of adults (≥ 18 years) residing in the eight participating countries or regions. The participants were randomly selected from online panels, curated by the polling firms using various sources. Substantial efforts were made to maximize census representation. In addition to using targeted recruitment to ensure inclusion of hard-to-reach groups, two strategies ensured the optimal representativeness of the sample: the use of software generating representative samples of the population (i.e., quotas sampling) and weighting of data based on age, sex and region in each participating country or region. For additional information concerning the recruitment procedure, the published article on the first phase of the study may be consulted [22]. The final sample of each measurement wave consisted of about 1000 participants for each of the countries or regions under study (with the exception of Canada, which was oversampled, making up a total of 8000–9000 participants per wave. The oversampling in Canada aimed to better analyze the situation in Quebec (the only province that is primarily francophone), as it differs greatly from the rest of Canada in terms of pandemic epidemiology, culture, political views, as well as official language.

### 2.3. Data Collection

An online questionnaire was used as the data collection instrument. The questionnaire is centred on the concepts of the Knowledge–Attitude–Practice (KAP) model [23], allowing the exploration of diverse themes such as risk perceptions and beliefs, positive and negative attitudes, as well as adaptive and maladaptive behaviours [18]. Data on sociodemographic characteristics was also collected, to perform more specific analyses. The questionnaire was revised following the first data collection with the collaboration of international partners to better explore topics that demonstrated significant importance in the first wave. These modifications remained minor and did not compromise the comparability between phases of the study. All the questions remained closed-ended and the average time of completion remained under 20 min. The questionnaire was translated into English, French, German, Flemish, Italian and Chinese and then validated by the project collaborators fluent in those languages.

### 2.4. Psychological Outcomes

Probable generalized anxiety (GAD) and major depressive episodes (MDE) were conserved as the two main assessed psychological outcomes. To evaluate these outcomes, the online questionnaire contained the GAD-7 and the Patient Health Questionnaire-9 (PHQ-9) scales, based on the diagnostic criteria for GAD and MDE described in DSM-IV. Although these scales are designed to be used in clinical settings by professionals, their use is also appropriate in population-based studies and they are often used as such. For each item on the respective scales, participants were asked to state their frequency according to four options (not at all, several days, over half the days or nearly every day). While the GAD-7 score ranges between 0–21 and the PHQ-9 score ranges from 0–27, the cut-off score of 10 or above for both scales identifies moderate to severe symptoms of GAD or MDE [24,25]. These scores signify probable GAD or MDE levels that require professional evaluation. These variables were maintained as the main psychological outcomes, to explore their evolution among the first and second wave of the pandemic.

### 2.5. Multilevel Variables

In the first phase of the study, factors that were assessed in the online survey were primarily factors that positively or negatively influenced the psychological response to the pandemic at an individual level. They were classified into four distinct categories, that is demographic characteristics, factors related to the pandemic, factors related to the infodemic, and individual psychological resources. These same factors were also explored in the second phase of the study (listed in Appendix A). In addition to these variables, the second phase of our study explores the effect of country-level variables on the psychological response to the pandemic. These factors include economic, geographic, social and COVID-19-related factors (listed in Appendix B). The latest national statistics available were retrieved from the governmental websites of each respective country and from nongovernmental organization databases. This information was then compiled in order to be used in the country-level analyses.

### 2.6. Sociodemographic Variables

The majority of the sociodemographic variables assessed remained unchanged from the ones used in the first phase of the project. These are sex (female, male) and household composition (living alone, living with others including children, living with others but without children). Being an essential worker was also assessed; however, this characteristic was further broken down to distinguish healthcare and social service workers from other essential workers (e.g., law enforcement, emergency services, provider of essential goods, educational institution). Similar to the previous phase, education level was not included in the international analyses. A barrier encountered in the previous analyses was the varying education systems among participating countries/regions, making it difficult to assess degree equivalency. To counter this, the data collected was converted to corresponding International Standard Classification of Education [26]. However, the question used to collect highest attained education in our international survey was not adapted to this classification, making it difficult to categorize the education level appropriately. As this posed a risk of introducing an information bias, this variable was not considered in subsequent analyses.

### 2.7. Data Analysis

The STrengthening the Reporting of OBservational studies in Epidemiology (STROBE) guidelines were used to inform the reporting of analyses [27]. Initially, participant characteristics were described, partitioned by the participating countries and measurement waves (either June or November 2020). A corrected weighted Pearson χ^2^ test was used to compare these participant characteristics across countries, by measurement wave. Treating countries as fixed effects, binomial regression models (with identity link function) were used to estimate rates of GAD or MDE indication by sex, age groups, country, measurement waves and two-factor interactions. As participants were nested within countries, and individual-level and country-level variables were available for analysis, a multilevel mixed-effects logistic regression model was considered and compared against the fixed effects binomial regression model, using the Bayesian Information Criterion (BIC). The BIC is used to select between these competing models; it rewards for goodness-of-fit to the data but penalizes for model complexity, with the preferred model balancing these opposing demands and yielding the lowest BIC statistic [28]. The multilevel model treated countries as random intercept effects with participants nested with them. Next, crude analyses were conducted, individually exploring sociodemographic and potential stressor variables and their interaction over the measurement waves, after adjusting for sex, age groups, country, measurement waves and their significant two-factor interactions. In the spirit of Sun and colleagues [29], all main effect variables were utilized in pursuant multivariable models without selection, although only significant interaction terms were entertained. Two multivariable models were considered, as the Hong Kong survey included a subset of variables (i.e., false beliefs score not available for this region); one including Hong Kong participants and the second excluding them. All analyses were conducted using Stata SE version 16.0 (StataCorp, College Station, TX, USA), accommodated the survey sampling weights, and two-tailed α = 0.05 defined significance.

## 3. Results

### 3.1. Participants and Their Characteristics

The final sample totaled 17,833 adults; 8806 from measurement wave 1 (June 2020) and 9027 from wave 2 (November 2020). To ensure the representativeness of the sample, the data were weighted according to sex, age and region of residence. When observing the sample as a whole, 51.8% of the participants were females, 49.2% were aged from 18–44 years old, 30.2% lived in households with children and 26.0% stated that they were essential workers, of whom 36.5% were healthcare or social service workers. Table 1 details the sociodemographic characteristics of the participants for each country for each wave.

When analyzing the data distribution at each measurement wave for sex, age, household composition and essential worker status using the corrected weighted Pearson χ^2^ test, significant differences were observed among countries for all characteristics (*p* < 0.001) with the exception of sex (*p* = 0.68 and *p* = 0.70 for June and November, respectively). When comparing these values to the national statistics of each country, the study samples appeared broadly representative. In addition, the sample characteristics resembled closely the sample previously obtained during the first phase of the study, which allowed a more accurate comparison between the two waves.

### 3.2. Psychological Outcomes

Overall, the data demonstrated an increase in negative psychological outcomes between the first and second measurement waves. Probable GAD was indicated by 21.0% and 23.6% of participants in June and November, respectively, an increase that was significant (corrected weighted Pearson χ^2^ test *p* < 0.001). Similarly, probable MDE was indicated by 25.5% and 27.8% of participants in June and November, showing a significant increase (corrected weighted Pearson χ^2^ test *p* = 0.002). For measurement wave 1 (June 2020), 1427.7 (16.2%) participants were indicated for both probable GAD and MDE and 2660.6 (30.2%) were indicated for either probable GAD or MDE; whereas for the second measurement wave, 1700.8 (18.8%) participants were indicated for both probable GAD and MDE and 2937.3 (32.5%) were indicated for either. More information concerning the indication of GAD/MDE can be found in Appendix C. With the exception of the USA, all countries had higher indication rates for observed probable GAD or MDE in June than November 2020, as depicted in Figure 1. This figure also shows substantial differences between countries, with rates in the USA, England, Hong Kong and the Philippines generally higher and those in Switzerland and Belgium generally lower. In fact, probable GAD or MDE indications ranged from 22.3% in Switzerland to 38.8% in the Philippines (in November 2020). In binomial regression models, these indications remained significantly different between countries even after adjusting for sex and age (*p* < 0.001).

When modelling weighted probable GAD or MDE indications, adjusting for sex, age, country, and measurement wave, together with considered two-factor interaction terms, the interaction between measurement wave × age was significant (*p* < 0.001) whereas the interactions between measurement wave × sex (*p* = 0.62), measurement wave × country (*p* = 0.30), and age × sex (*p* = 0.20) were not. This implies that the pattern of indications over age significantly changed between the measurement waves. Repeating the binomial regression model analysis, keeping the significant measurement wave × age interaction term, revealed that indications were significantly different between countries (*p* < 0.001), sex (*p* < 0.001), age (*p* < 0.001), measurement waves (*p* < 0.001), and measurement wave × age (*p* < 0.001). Figure 2 depicts the estimated weighted proportion of participants with probable GAD or MDE indications derived from this model. As can be observed from this figure, rates among female, younger, and measurement wave 2 (November 2020) participants are generally higher than males, older and measurement wave 1 (June 2020) participants. Note also the generally higher changes between June and November rates in the younger age groups compared to the older age groups; see Figure 2.

### 3.3. Crude Multilevel Mixed-Effects Models

In the analysis including sex, age, measurement wave, and the measurement wave×age interaction, the intercept-only multilevel mixed-effects model of probable GAD or MDE indication was superior to the model treating countries as fixed effects (BIC: 20,623.0 vs. 20,715.1, respectively). Thus, pursuant analyses treated participants as being nested within countries, which were modelled as random intercepts. In this multilevel mixed-effects model, females had odds of probable GAD or MDE indication 1.24 (95% CI: 1.09, 1.42) compared to that of males, and to those aged ≥ 65 years; participants aged 18–24 years had odds of probable GAD or MDE indication 4.22 (95% CI: 2.72, 6.53) higher in June and 7.93 (95% CI: 4.55, 13.8) higher in November. The estimated intraclass correlation (ICC) among participants within the same country was 0.020 (95% CI: 0.010, 0.038). In other words, only 2% of the variance in psychological outcomes under study were explained by country-level factors, suggesting that most of the variation lies between people. Table 2 includes the estimated odds of probable GAD or MDE indication by measurement wave for these sex and age group variables.

The considered sociodemographic and potential stressor variables were next individually added to this model, together with their interaction over measurement wave (see Table 2). Variables measuring household composition, financial loss, threat perceived for oneself and/or family, false beliefs score, friend/family/coworkers as a regular source of information, and sense of coherence were each significantly associated with probable GAD or MDE indication and their associations had significant interactions by measurement wave (all *p* < 0.05). Conversely, variables measuring essential worker status, self-isolation/quarantine, threat perceived for country and/or world, being a victim of stigma, level of information about COVID-19, trust in authorities score, and social networks used as a regular source of information had significant associations with probable GAD or MDE indication but no such interaction with measurement wave.

Next, country-level predictors were considered. Table 3 presents the considered country-level variables, together with crude multilevel mixed-effects logistic model estimates, adjusted for sex, age, measurement wave, and the measurement wave × age interaction. Only the total population numbers (*p* < 0.001) and the GINI index (*p* = 0.03) were significantly related to probable GAD or MDE indication.

### 3.4. Multivariable Multilevel Mixed-Effects Models

Variables and interactions that were significant within the crude models were next considered simultaneously together in a multivariable model, without variable selection. As false beliefs score was not elicited from Hong Kong participants, two analyses were conducted (1) including all countries omitting the false beliefs score variable; (2) including all variables and omitting Hong Kong participants. Table 4 includes the adjusted estimates derived from these models. For the analysis (1), all included interactions with measurement wave remained significant except for household composition (*p* = 0.14), and all considered main effects also remained significant except for the country-level GINI index (*p* = 0.56) and the individual-level variable friend/family/coworkers as a regular source of information (*p* = 0.32), although the interaction term for this latter variable was significant (*p* = 0.04). Notably, in this analysis, those aged 18–24 years had high-adjusted odds for probable GAD or MDE indication (2.73; 95% CI: 2.03, 3.68) compared to their older aged ≥ 65 years, and this was worse in November (4.23; 95% CI: 2.58, 6.93). Additionally, a weak SOC yielded a high-adjusted odd for probable GAD or MDE indication at measurement wave 1 (3.17; 95% CI: 2.69, 3.73), which significantly increased in November (3.95; 95% CI: 3.37, 4.62; *p* = 0.008) (see Table 4).

Undertaking analysis (2), generally similar patterns emerged. All included interactions with measurement wave remained significant except for household composition (*p* = 0.25), threat perceived for oneself and/or family (*p* = 0.09), false beliefs score (*p* = 0.93), and friend/family/coworkers as a regular source of information (*p* = 0.22), and all considered main effects also remained significant except for the country-level GINI index (*p* = 0.18) and the individual-level variable friend/family/coworkers as a regular source of information (*p* = 0.75). Again, those aged 18–24 years and weak SOC carried a relatively high odds of probable GAD or MDE indication compared to their peers—a burden that significantly worsened in November.

## 4. Discussion

The second phase of our interdisciplinary and international survey points to a multitude of findings that contribute to enhance our current understanding of the mental health crisis amid the pandemic. First, large and persistent psychosocial impacts of the COVID-19 among adults were found in a set of very diverse countries (in terms of epidemiological situations and sociocultural backgrounds) all over the world. Second, by using the exact same methodology (e.g., same target populations, recruitment strategies and measurement tools), worsening of anxiety and depression levels have been documented, particularly in young adults, between Phase 1 (May–June 2020) and Phase 2 (November 2020). Third, beyond young age, a wide range of factors negatively influencing mental health in times of the pandemic were highlighted, important factors (based on the magnitude of the effect size) were found, including a weaker SOC, COVID-19-related false beliefs, and self-isolation or quarantine.

Overall, we found a small but significant deterioration with the progression of the COVID-19 pandemic in the psychological health of the adult population from eight countries and four continents. A global increase of 7.9% in the indication of either GAD or MDE was indeed noted from June–November 2020, with almost a third of the participants (32.5%) exhibiting symptoms consistent with one of these mental disorders in November. As discussed in an earlier article, such prevalence is much greater than what was estimated in the prepandemic era [22].

One key observation in the current study is of great importance and deserves full attention: in all countries but Switzerland, more than half of young adults reported symptoms consistent with GAD or MDE. This age group, which was already significantly affected by the pandemic in June 2020, showed deteriorating psychological health at a faster pace in the second half of 2020 than any other age group. There are many possible explanations for these disturbing results. Regardless of the pandemic, the transition from adolescence to adulthood has always been a stressful period filled with changes and adjustments [30]. Since the beginning of the pandemic, youth faced additional stressors, including an overwhelming sense of loneliness; a reduction in social [31], sport and cultural activities, homeschooling; loss of employment and financial stress for many [32]; feeling of injustice due to the imbalance between efforts required and the rewards; and larger effects of the infodemic on this age group (resulting in more confusion and anger) [32]. At first thought older adults may seem to be more vulnerable to negative psychological outcomes; however, this group was found to be less affected. These results were echoed in other studies were despite their high percentage of emotional distress, adults aged 60 years and older remained at a lower risk of developing depressive and stress consequences from COVID-19 and lockdown than their younger counterparts [33].

Interestingly, the evolution of the epidemiological situation regarding the COVID-19 morbidity and mortality at the country-level does not seem to influence rates of probable anxiety or mood disorders. Indeed, despite being strongly affected by the second wave of COVID-19 during the time of the study, Switzerland and Belgium remain the two countries with the lowest indication for negative psychological outcomes. These countries seem to be less affected than New Zealand and Hong Kong, which reported very few new cases of the virus at the beginning of the month of November. Several studies have exposed the negative impacts of public health measures such as lockdown or stay-at-home recommendations on mental health regardless of the epidemiological situation in a country [34]. The different measures applied by each government, their level of severity, their varying length of implementation, and the way the population receives, understands and perceives these measures could explain the variances in mental health outcomes between the participating countries [34]. The implementation of regional-based measures and lockdowns seen in both Canada and the United States can act as a mental health stressor due to unclear information from public health authorities. This coincides with the observed negative psychological outcomes in these two countries. During the SARS epidemic in Toronto (Canada) in 2003, confusion stemmed from the content of various public health messages due to poor coordination between jurisdictions and levels of government [34]. Then, it is suggested that longer quarantines could be linked to poorer mental health [34]. The length of the lockdown enforced in the Philippines, spanning nearly six months since the start of the pandemic, is reflected in the results of the study as this country has one of the highest levels of GAD and MDE observed [35]. Finally, lockdowns in Belgium and Switzerland were more relaxed than the other participating countries, with establishments such as businesses and restaurants remaining open (with restrictions). These elements could contribute to maintaining a relative normalcy, aiding in mitigating the psychological impacts of the pandemic.

When specifically considering the lower prevalence of GAD and MDE observed for Belgium and Switzerland, many potential factors may play a role in these findings. One aspect that must be considered is the timing between the waves (periods of confinement and deconfinement) in each respective country and the data collection periods. The first phase of data collection in June coincided with the first deconfinement in Belgium and the second phase in November fell just at the beginning of the second confinement. Another study done by the University of Louvain in Belgium revealed high levels of psychological distress for the month of March (the start of the confinement) and April (the peak of the first wave) [36]. Another important aspect to consider is the internal geopolitical context of each country. Several participating countries have volatile political situations. Although our questionnaire sought to evaluate changes in the mental health outcomes caused by the pandemic, it is difficult for participants to disassociate from co-occurrent events that can also affect their mental health. For instance, the USA is the only participating country that experienced a decline in anxiety and depression levels. This could be owed to the fact that the USA presidential election day was on 3 November 2020, a few days before the second phase of data collection began.

Furthermore, the total population size of a country was found to act as independent predictor of either probable GAD or MDE. At first glance, it could be hypothesized that a larger size country could be affected negatively by both complex crisis management and communication strategies, resulting in increased negative mental health outcomes related to the pandemic. However, the relationship at stake might not be directly attributable to the population size, but rather their impact on trust, confidence and perception of proximity with authorities/elites (including health authorities in the case of COVID-19). Confidence and trust in national and/or local state authorities are expected to be lower in larger populations, often facing more inequalities and economic stress [37,38]. Lower trust in authorities leads to a greater sense of uncertainty, stress about future outcomes of the pandemic, and more dis/misinformation (as less confidence towards authorities incites the consumption and sharing of lower quality information), which can all be detrimental to mental health outcomes in a population [39]. As seen in this study, high scores related to false beliefs and low scores related to trust in authorities both increase negative mental health outcomes, which corroborates the points raised above. In fact, infodemic-related factors (including mistrust, confusion and false beliefs, overload of information, and the use of social media to become informed) were found in our study to play an equally important role, if not more important, than pandemic-related factors in explaining psychological health in times of pandemic. This strongly supports the need for further interdisciplinary studies to investigate a comprehensive range of traditional and less traditional factors. This is crucial for a better understanding of what really undermines mental health in this unique era where erroneous information is spreading even more quickly than the SARS-CoV-2 itself.

As observed in the first phase of the study, the SOC is still critical in protecting against adversity caused by the two concurrent crises (i.e., the pandemic and the infodemic). Recent work has identified courses of action shown to increase SOC and, in general, the adaptive capacities of individuals and communities during stressful situations such as the ongoing pandemic [40]. Interventions in health promotion that aim to support the strengthening of SOC need to focus on empowerment and reflection, and should be guided by various principles (i.e., positive approach, collaborative work, locally based, adapted to the context and local culture, inclusiveness). Such interventions can take several forms, going from programs focused on the development of mindfulness to artistic and cultural activities.

This study has several limitations, the main one being its cross-sectional nature, precluding our capacity to infer a causal link between risk/protective factors and mental health outcomes. When interpreting the changes in the data over time, it is important to consider that this study uses a repeated-cross sectional design and not a longitudinal approach, making it more difficult to attribute the changes in negative psychological outcomes to the various variables studied. Additionally, despite tremendous efforts in achieving representativeness in our sample, some groups of the population may be underrepresented, including adults with lower literacy levels and those not having access to a computer or the Internet. When considering the possible distorting effects of the weighting of the data, the weights had minimal effects on our main estimates (e.g., probable GAD or MDE). Indeed, probable GAD or MDE was indicated by 32.2% and 32.5% of the respondents using the unweighted and the weighted data, respectively (a 0.9% difference), suggesting that weights did not led to distortions of effects. Although the use of a cut-off value in the PHQ-9 and GAD-7 scales facilitates the interpretation of the data relating to the GAD or MDE (probable disorder: yes or no), it can also lead to some limits. As the original data takes a continuous from, some details are lost when using a cut-off and the prevalence of some of the factors may be overestimated [41]. Another limitation would be the absence of a valid education level measure (or another indicator of individual socioeconomic status) suitable for the international analyses. Although the lack of such a variable is unfortunate, it was expected as participating countries were purposely selected for their diversity in terms of sociocultural backgrounds. In an attempt to minimize this gap, country-level variables were considered in phase 2 of the international survey, including variables related to the economic context (e.g., GINI index) and others related to the social determinants of health (e.g., mean age of schooling). Although various country-level data were considered in our analyses, including COVID-19 epidemiological data, the present study did not account for changes in public messaging, implemented measures and risk over time. These were difficult to assess as these variables differed greatly at a national level. For example, in Canada, during the observed periods, each province implemented their own restriction guidelines with the federal government only providing general recommendations such as social distancing, face masks and stay at home recommendation as well as enforcing a quarantine for travelers [42]. Other countries had different approaches, such as England, which implemented a strict lockdown at a national level for a few weeks in November 2020 before returning to a regional approach [43].

## 5. Conclusions

The world has entered into a new era dominated by psychological suffering and rising demand for mental health interventions along a continuum from health promotion to specialized healthcare. More than ever, there is a need to foster individual and collective resilience. Each community should attempt to build and share a common vision of its problems and vulnerabilities, but also of its resources and capacities, in order to jointly develop solutions tailored to the local context and to thrive in these uncertain times. As observed, poor mental health outcomes associated with lockdown and other collateral damages of the pandemic (and the infodemic) could be better mitigated with ongoing assessment of psychosocial impacts and associated stressors, informed decisions and thoughtful interventions at the individual and collective levels [44].

There is still so much research needed in the practical and political fields. With regard to our research project, new questions were added to the measurement tool of the second phase (November 2020), including additional psychosocial outcomes (e.g., serious suicidal ideation, domestic violence, alcohol and cannabis consumption, physical activity level) as well as another potential factors that may influence these outcomes in times of pandemic, that is personal political leaning (centrist, left-leaning, right-leaning). As a next step, our interdisciplinary team will explore how the pandemic may have influenced these very important public health issues and the contributing role of this “new” factor that has been rarely, if ever, assessed in times of pandemic.

## Figures and Tables

**Figure 1 ijerph-18-04845-f001:**
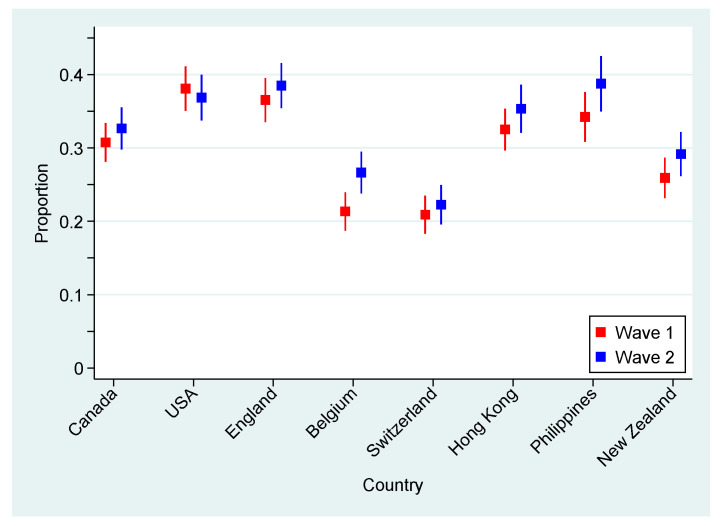
Estimated weighted proportion of participants indicated for either probable GAD or MDE by countries or regions, measurement waves 1 (June 2020) and 2 (November 2020).

**Figure 2 ijerph-18-04845-f002:**
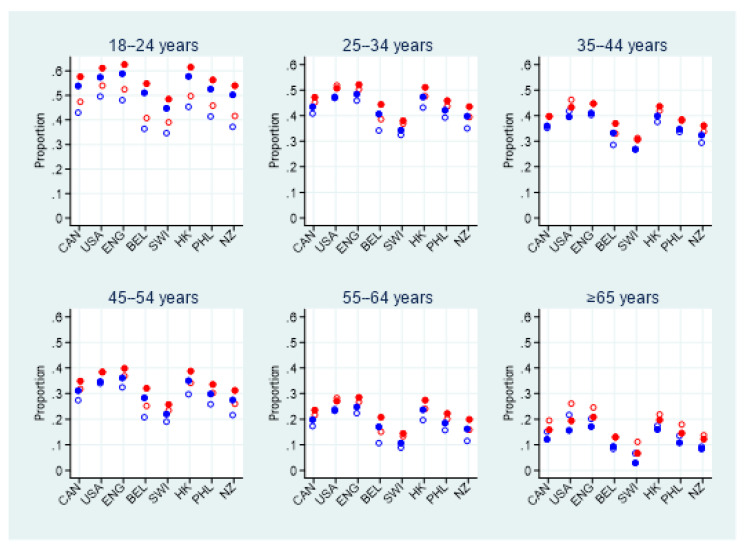
Estimated weighted proportion of participants indicated for either probable GAD or MDE from the binomial regression model adjusted by age, sex, countries, measurement wave, and the measurement wave × age interaction presented by country, partitioned into age groups. Females are denoted by red circles, males by blue circles, measurement wave 1 (June 2020) with hollow circles and measurement wave 2 (November 2020) with solid circles.

**Table 1 ijerph-18-04845-t001:** Weighted distribution of participants’ demographic characteristics by country and measurement wave (June or November 2020).

		Canada		United States		England		Belgium		Switzerland		Hong Kong		Philippines		New Zealand	
Meas. wave		1	2	1	2	1	2	1	2	1	2	1	2	1	2	1	2
		%	%	%	%	%	%	%	%	%	%	%	%	%	%	%	%
Numbers		1501	2004	1065	1003	1041	1000	1015	1014	1002	1000	1140	1002	1041	1003	1001	1001
Sex ^a^																	
	Female	48.4	48.3	48.5	48.1	48.8	47.8	48.6	48.4	47.7	47.8	45.1	45.0	49.2	49.3	48.6	48.6
	Male	51.6	51.7	51.5	51.9	51.2	51.2	51.4	51.6	52.3	52.2	54.9	55.0	50.6	50.7	51.4	51.4
Age (years)																	
	18–24	10.9	10.9	5.5	8.0	11.1	11.1	6.2	5.6	9.5	9.5	9.5	9.5	21.6	22.6	12.2	12.2
	25–34	16.4	16.4	21.2	18.7	17.4	17.4	20.5	21.2	14.4	14.4	17.2	17.2	25.0	25.0	18.4	18.4
	35–44	16.2	16.2	17.9	17.9	16.3	16.3	13.7	11.6	13.8	13.8	18.1	18.1	20.1	20.1	16.3	16.3
	45–54	17.9	17.9	19.1	19.1	17.9	17.9	20.7	22.5	17.6	17.6	19.1	19.1	15.5	15.5	17.5	17.5
	55–64	17.5	17.5	17.8	17.8	14.5	14.5	16.9	15.9	23.9	17.1	17.7	17.7	10.2	12.5	15.7	15.7
	≥ 65	21.1	21.1	18.4	18.4	22.8	22.8	22.0	23.2	20.8	27.5	18.4	18.4	7.7	5.3	19.9	19.9
Household composition ^b^																	
	Alone	20.2	18.3	21.9	22.9	20.7	21.2	18.9	18.8	23.6	27.1	6.5	6.7	4.8	3.4	18.0	15.6
	With children	25.1	22.4	32.2	30.2	27.9	25.7	22.2	22.2	22.1	18.5	31.0	27.8	52.9	55.0	32.6	34.0
	With others	54.7	59.3	45.9	46.9	51.4	53.2	58.9	59.0	54.3	54.4	62.5	65.5	42.3	41.6	49.4	50.4
Essential worker ^c^																	
	No	75.9	73.6	78.2	73.0	73.1	73.9	82.2	77.4	77.2	78.3	64.3	61.3	81.2	70.4	72.9	74.3
	Yes: health	6.8	8.9	7.8	9.2	9.5	7.9	7.5	6.1	13.0	9.8	13.6	10.9	8.1	10.9	9.9	8.9
	Yes: other	17.3	17.5	14.1	17.8	17.3	18.2	10.2	16.5	9.8	11.9	22.1	27.8	10.7	18.7	17.2	16.8

Note: ^a^ 25 participants at measurement wave 1 and 42 participants at measurement wave 2 did not identify with female or male gender, or preferred not to answer this question, so had their sex set to missing; ^b^ 2 participants at each measurement wave had missing or invalid data; and, ^c^ 180 and 182 participants at measurement waves 1 and 2 had missing or invalid data.

**Table 2 ijerph-18-04845-t002:** Distribution of weighted probable GAD or MDE indications for potential risk and protective factors, together with estimated crude multilevel mixed-effects logistic model odds ratios (ORs) and associated 95% confidence intervals (CIs), adjusted for sex, age, measurement wave, and the measurement wave × age interaction.

			GAD or MDE		Wave 1 (June 2020)		Wave 2 (November 2020)	
		N	n	(%)	OR	(95% CI)	OR	(95% CI)
Demographic characteristics								
Sex								
	Female	9222.9	3067.1	(33.3)	1.24	(1.09, 1.42)	1.24	(1.09, 1.42)
	Male	8543.1	2497.6	(29.2)	1	(reference)	1	(reference)
*Age (years)*								
	18–24	1946.0	972.4	(50.0)	4.22	(2.72, 6.53)	7.93	(4.55, 13.8)
	25–34	3302.8	1444.6	(43.7)	3.87	(2.58, 5.82)	5.23	(3.22, 8.50)
	35–44	2927.1	1098.3	(37.5)	3.05	(2.04, 4.57)	3.79	(2.42, 5.94)
	45–54	3257.9	1001.1	(30.7)	2.11	(1.59, 2.79)	3.09	(2.13, 4.47)
	55–64	2945.9	566.6	(19.2)	1.14	(0.76, 1.72)	1.64	(1.11, 2.43)
	≥ 65	3453.2	514.9	(14.9)	1	(reference)	1	(reference)
Household composition								
	Alone	3010.2	837.4	(27.8)	1	(reference)	1	(reference)
	With children	5275.6	2063.6	(39.1)	1.17	(0.92, 1.48)	1.02	(0.81, 1.28)
	With others	9543.3	2695.8	(28.2)	0.90	(0.82, 0.99)	0.84	(0.74, 0.96)
Essential worker								
	No	12,958.1	3688.5	(28.5)	1	(reference)	1	(reference)
	Yes: health	1610.5	646.3	(40.1)	1.32	(1.05, 1.66)	1.32 *	(1.05, 1.66)
	Yes: other	2902.5	1121.1	(38.6)	1.28	(1.14, 1.45)	1.28 *	(1.14, 1.45)
Factors related to the pandemic								
Self-isolation/quarantine								
	No	7785.7	1870.7	(24.0)	1	(reference)	1	(reference)
	Yes, case/symptoms-free	7415.3	2374.2	(32.0)	1.49	(1.42, 1.58)	1.49 *	(1.42, 1.58)
	Yes, case or symptoms	2124.9	1123.6	(52.9)	2.78	(2.14, 3.62)	2.78 *	(2.14, 3.62)
Financial losses								
	No	8525.7	1996.5	(23.4)	1	(reference)	1	(reference)
	Yes	8346.6	3223.3	(38.6)	1.73	(1.59, 1.88)	2.10	(1.83, 2.43)
	Unsure/unknown	960.7	378.2	(39.4)	1.80	(1.52, 2.14)	1.90	(1.36, 2.64)
Threat perceived for oneself and/or family								
	High	6387.0	2734.0	(42.8)	2.51	(2.19, 2.87)	2.18	(1.89, 2.52)
	Otherwise	10,904.0	2689.1	(24.7)	1	(reference)	1	(reference)
Threat perceived for country and/or world								
	High	12,775.8	4253.5	(33.3)	1.50	(1.32, 1.71)	1.50 *	(1.32, 1.71)
	Otherwise	4464.2	1160.2	(26.0)	1	(reference)	1	(reference)
Being a victim of stigma								
	No	14,104.7	3810.5	(27.0)	1	(reference)	1	(reference)
	Yes	2362.2	1274.7	(54.0)	2.66	(2.15, 3.29)	2.66 *	(2.15, 3.29)
	Decline to answer	1366.2	512.8	(37.5)	1.46	(1.22, 1.75)	1.46	(1.22, 1.75)
Factors related to the infodemic								
Level of information about COVID-19								
	High (9–10)	5577.7	1787.8	(32.1)	1	(reference)	1	(reference)
	Otherwise (1–8)	12,255.3	3810.2	(31.1)	0.91	(0.87, 0.96)	0.91 *	(0.87, 0.96)
Trust in authorities score								
	Q1 (low)	4716.5	1713.3	(36.3)	1.39	(1.09, 1.78)	1.39 *	(1.09, 1.78)
	Q2	4103.4	1333.3	(32.5)	1.23	(1.09, 1.38)	1.23 *	(1.09, 1.38)
	Q3	4422.8	1277.7	(28.9)	1.08	(0.95, 1.23)	1.08 *	(0.95, 1.23)
	Q4 (high)	4590.3	1273.6	(27.7)	1	(reference)	1	(reference)
False beliefs score †								
	Q1 (low)	4139.5	843.3	(20.4)	1	(reference)	1	(reference)
	Q2	3810.6	927.4	(24.3)	1.23	(1.08, 1.40)	1.25	(1.13, 1.37)
	Q3	3932.1	1237.3	(31.5)	1.69	(1.21, 2.36)	1.72	(1.39, 2.14)
	Q4 (high)	3808.8	1865.0	(49.0)	3.26	(2.31, 4.61)	2.83	(2.10, 3.83)
Social networks used as a regular source of information								
	Often/always	5336.9	2200.1	(41.2)	1.44	(1.17, 1.78)	1.44 *	(1.17, 1.78)
	Sometimes/never	11,771.1	3191.8	(27.1)	1	(reference)	1	(reference)
Friend/family/coworkers as a regular source of information								
	Often/always	7266.6	2554.1	(35.1)	1.31	(1.17, 1.46)	1.10	(0.95, 1.27)
	Sometimes/never	10,204.4	2918.2	(28.6)	1	(reference)	1	(reference)
Individual psychological resources								
Sense of coherence								
	Strong (5–6)	5368.9	689.5	(12.8)	1	(reference)	1	(reference)
	Weak (0–4)	12,464.1	4908.5	(39.4)	3.26	(2.80, 3.79)	4.18	(3.48, 5.03)

† Data not collected in Hong Kong. * There was no significant interaction with measurement wave.

**Table 3 ijerph-18-04845-t003:** Country-level variables, together with estimated crude multilevel mixed-effects logistic model odds ratios (ORs) and associated 95% confidence intervals (CIs) of probable GAD or MDE indications, adjusted for sex, age, measurement wave, and the measurement wave × age interaction.

	Canada	USA	England	Belgium	Switzerland	Hong Kong	Philippines	NZ	OR	(95% CI)
Economic context										
GDP	46,194.70	65,118.40	42,300.30	46,116.70	81,993.70	48,755.80	3485.10	42,084.40	† 0.98	(0.90, 1.07)
Human Development Index	0.922	0.920	0.920	0.919	0.946	0.939	0.712	0.921	0.88	(0.26, 2.98)
Unemployment rate	8.9%	6.7%	4.8%	5.1%	3.2%	6.4%	8.7%	5.3%	1.06	(0.98, 1.15)
GINI index	32.1	45.0	32.4	25.9	29.5	53.9	44.4	36.2	1.02	(1.00, 1.03)
Geographic context										
Total population (million)	37.06	326.69	56.20	11.43	8.51	7.45	106.65	4.84	‡ 1.16	(1.07, 1.26)
Population growth	1.4	0.5	0.55	0.5	0.7	0.8	1.4	1.0	0.94	(0.59, 1.50)
Population density	4.1	35.7	274.7	377.4	215.5	7096.2	357.7	18.4	1.00	(0.99, 1.01)
Social determinants of health										
Life expectancy	82	79	81	82	84	85	71	82	0.99	(0.96, 1.02)
Median age (years)	41.8	38.5	40.6	41.6	42.7	45.6	24.1	37.2	1.00	(0.99, 1.01)
Mean years of schooling	13.3	13.4	13.0	11.8	13.4	12	9.4	12.7	1.02	(0.92, 1.14)
COVID-19 epidemiological data per 100,000										
Cumulative cases—June/November	245/737	550/3,206	393/2,917	516/4,648	360/2,975	15/73	17/382	24/41	1.00	(1.00, 1.00)
Cumulative deaths—June/November	19.6/28.4	32.0/73.9	60.6/115.6	82.0/125.1	19.4/34.2	0.05/1.45	0.89/7.34	0.46/0.52	1.00	(0.99, 1.01)
Cumulative tests *—June/November	4.60/27.41	6.07/49.43	2.53/62.63	8.00/43.74	4.75/27.20	3.20/100.59	0.33/4.94	5.82/24.35	1.00	(0.99, 1.01)

* as a % of the total population; † estimated on GDP/10,000; ‡ estimated on total population (100 million).

**Table 4 ijerph-18-04845-t004:** Multivariable multilevel mixed-effects logistic model odds ratios (ORs) and associated 95% confidence intervals (CIs) of probable GAD or MDE indications.

		(1) Including Hong Kong (*n* = 15,863)				(2) Excluding Hong Kong (*n* = 13,877)			
		Wave 1 (June 2020)		Wave 2 (November 2020)		Wave 1 (June 2020)		Wave 2 (November 2020)	
		OR	(95% CI)	OR	(95% CI)	OR	(95% CI)	OR	(95% CI)
Demographic characteristics									
Sex									
Female		1.25	(1.11, 1.41)	1.25	(1.11, 1.41)	1.36	(1.27, 1.44)	1.36	(1.27, 1.44)
Male		1	(reference)	1	(reference)	1	(reference)	1	(reference)
Age (years)									
18–24		2.73	(2.03, 3.68)	4.23	(2.58, 6.93)	3.11	(2.62, 3.71)	5.04	(3.39, 7.48)
25–34		2.42	(1.78, 3.29)	2.90	(1.93, 4.35)	2.57	(2.01, 3.28)	3.26	(2.29, 4.65)
35–44		1.96	(1.47, 2.62)	2.26	(1.59, 3.23)	2.20	(1.83, 2.64)	2.56	(2.00, 3.27)
45–54		1.65	(1.29, 2.11)	1.98	(1.46, 2.70)	1.88	(1.60, 2.22)	2.17	(1.59, 2.97)
55–64		1.12	(0.77, 1.63)	1.36	(0.95, 1.93)	1.35	(1.11, 1.65)	1.45	(1.01, 2.09)
≥65		1	(reference)	1	(reference)	1	(reference)	1	(reference)
Household composition									
Alone		1	(reference)	1	(reference)	1	(reference)	1	(reference)
With children		1.25	(1.00, 1.55)	1.06	(0.90, 1.23)	1.12	(0.94, 1.34)	0.94	(0.82, 1.08)
With others		0.93	(0.83, 1.05)	0.89	(0.81, 0.97)	0.92	(0.82, 1.02)	0.86	(0.81, 0.92)
Essential worker									
No		1	(reference)	1	(reference)	1	(reference)	1	(reference)
Yes: health		1.16	(0.91, 1.47)	1.16	(0.91, 1.47)	0.99	(0.86, 1.14)	0.99	(0.86, 1.14)
Yes: other		1.21	(1.13, 1.29)	1.21	(1.13, 1.29)	1.15	(1.07, 1.24)	1.15	(1.07, 1.24)
Factors related to the pandemic									
Self-isolation/quarantine									
No		1	(reference)	1	(reference)	1	(reference)	1	(reference)
Yes, case/symptoms-free		1.28	(1.19, 1.38)	1.28	(1.19, 1.38)	1.37	(1.30, 1.46)	1.37	(1.30, 1.46)
Yes, case or symptoms		2.02	(1.70, 2.39)	2.02	(1.70, 2.39)	2.01	(1.67, 2.42)	2.01	(1.67, 2.42)
Financial losses									
No		1	(reference)	1	(reference)	1	(reference)	1	(reference)
Yes		1.35	(1.18, 1.56)	1.59	(1.39, 1.82)	1.36	(1.13, 1.62)	1.55	(1.33, 1.81)
Unsure/unknown		1.62	(1.19, 2.21)	2.09	(1.54, 2.84)	1.55	(1.08, 2.23)	1.86	(1.34, 2.56)
Threat perceived for oneself and/or family									
High		2.14	(1.92, 2.39)	1.82	(1.56, 2.13)	2.01	(1.81, 2.24)	1.81	(1.52, 2.15)
Otherwise		1	(reference)	1	(reference)	1	(reference)	1	(reference)
Threat perceived for country and/or world									
High		1.21	(1.08, 1.35)	1.21	(1.08, 1.35)	1.28	(1.21, 1.36)	1.28	(1.21, 1.36)
Otherwise		1	(reference)	1	(reference)	1	(reference)	1	(reference)
Being a victim of stigma									
No		1	(reference)	1	(reference)	1	(reference)	1	(reference)
Yes		1.78	(1.54, 2.07)	1.78	(1.54, 2.07)	1.59	(1.41, 1.80)	1.59	(1.41, 1.80)
Decline to answer		1.17	(0.96, 1.42)	1.17	(0.96, 1.42)	1.06	(0.86, 1.31)	1.06	(0.86, 1.31)
Factors related to the infodemic									
Level of information about COVID-19									
High (9–10)		1	(reference)	1	(reference)	1	(reference)	1	(reference)
Otherwise (1–8)		0.85	(0.77, 0.95)	0.85	(0.77, 0.95)	0.84	(0.76, 0.93)	0.84	(0.76, 0.93)
Trust in authorities score									
Q1 (low)		1.68	(1.41, 2.00)	1.68	(1.41, 2.00)	1.51	(1.29, 1.77)	1.51	(1.29, 1.77)
Q2		1.36	(1.25, 1.48)	1.36	(1.25, 1.48)	1.26	(1.11, 1.43)	1.26	(1.11, 1.43)
Q3		1.16	(1.04, 1.28)	1.16	(1.04, 1.28)	1.12	(1.03, 1.23)	1.12	(1.03, 1.23)
Q4 (high)		1	(reference)	1	(reference)	1	(reference)	1	(reference)
False beliefs score *									
Q1 (low)		-	-	-	-	1	(reference)	1	(reference)
Q2		-	-	-	-	1.17	(1.07, 1.29)	1.15	(0.99, 1.33)
Q3		-	-	-	-	1.50	(1.12, 2.01)	1.56	(1.24, 1.95)
Q4 (high)		-	-	-	-	2.42	(1.78, 3.29)	2.33	(1.84, 2.95)
Social networks used as a regular source of information									
Often/always		1.27	(1.04, 1.54)	1.27	(1.04, 1.54)	1.24	(1.04, 1.48)	1.24	(1.04, 1.48)
Sometimes/never		1	(reference)	1	(reference)	1	(reference)	1	(reference)
Friend/family/coworkers as a regular source of information									
Often/always		1.06	(0.94, 1.19)	0.94	(0.80, 1.09)	0.98	(0.85, 1.13)	0.88	(0.74, 1.05)
Sometimes/never		1	(reference)	1	(reference)	1	(reference)	1	(reference)
Individual psychological resources									
Sense of coherence									
Strong (5–6)		1	(reference)	1	(reference)	1	(reference)	1	(reference)
Weak (0–4)		3.17	(2.69, 3.73)	3.95	(3.37, 4.62)	3.12	(2.71, 3.59)	3.89	(3.37, 4.49)
Country-level factors									
Total population (100 million)		1.16	(1.10, 1.22)	1.16	(1.10, 1.22)	1.31	(1.04, 1.64)	1.31	(1.04, 1.64)
GINI index		0.99	(0.98, 1.01)	0.99	(0.98, 1.01)	0.97	(0.92, 1.02)	0.97	(0.92, 1.02)

* Data not collected in Hong Kong.

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
