# Peer review of "The Evolution in Anxiety and Depression with the Progression of the Pandemic in Adult Populations from Eight Countries and Four Continents"

_ijerph, 2021, doi:10.3390/ijerph18094845_

Round 1

Reviewer 1 Report

In the article: “The Evolution in Anxiety and Depression with the Progression of the Pandemic in Adult Populations from Eight Countries and Four Continents” the authors present research on the consequences of the Covid-19 pandemic. They report that the mental health (i.e., anxiety depression) was worsening across time in nearly all of the 8 countries from 4 continents. Altogether this research is very impressive – as measures took place across time, countries and different age group. Also, the consideration of underlying reasons and coping factors is a promising approach. However, there are some concerns left regarding the presentation and discussion of the results – publishing a revised version of the paper would surely be an important contribution.

The reporting of the results needs to include more details. In some instances (e.g. page 8), the reader just gets the explanation that there was substantial difference and p-values – but there is no description which concrete test/analysis was undertaken to arrive at that result.

Even more important is the missing description regarding the meaning of the effects or interaction (e.g. page 11). The authors did only list the influential factors and interactions, without any description regarding the nature of the influence. If I understand for example the Figure 2 and the interaction of ageXwave correctly: the symptoms increase for younger people (who are generally scoring higher than older participants) and decrease in case of older people? Why is that? Even the higher score for younger participants left me puzzled – as the older ones are much more threatened by the pandemic. The only possibility of the lower scores for older participants that I can think of, is a self-selection bias. Nevertheless, the results should in any case be explained and also discussed in more detail.

Minor points:

  • Please include more details on the assessment of the dependent variables GAD/MDE. What was the response format of the PHQ and the GAD7? Include an item-example? Were the GAD/MDE scores averaged for the analyses or how was the variable conducted?
  • I was wondering if there are some copy-paste-mistakes for example in table 3 (the numbers regarding the waves of” self-isolation/quarantine” or e.g., regarding “being a victim of stigma” are exactly the same). Please check.

Reviewer 2 Report

The study examines prevalence of depression and anxiety symptoms in a large multi-national sample during April-November 2020. There are myriad studies using similar methods to characterize the impacts of COVID on mental health. While the authors take steps to ensure the data are somewhat representative of the population, there remain some distinct methodological shortcomings that are not fully acknowledged. The contribution of this study is also somewhat limited. The findings from the first wave have already been reported. Modest differences in prevalence are noted, although care needs to be taken in interpretation due to the use of repeated cross-sectional design. Otherwise the risk factors identified in the first paper (https://doi.org/10.3390/ijerph17228390) appear to be largely similar to those reported here. The paper could therefore be more concise and have a greater focus on what is new.

  1. Insufficient information about the weighting scheme is given. What was the range of weights? Furthermore, weighting only accounted for imbalances by sex, age and region - there is likely to be significant selection biases in that people with a personal interest in mental health are much more likely to complete a mental health survey, which is likely to considerably bias any prevalence estimates but was not directly accounted for by the weighting scheme. See, e.g., Pierce et al (2020). The Lancet Psychiatry, 7(7), 567-568. Consideration should be given to whether excessive (or minuscule) weights led to distortions of effects, and the limitations of weighting (e.g., https://doi.org/10.1177/096228029600500303, https://doi.org/10.1177/0049124194023002004, https://doi.org/10.1002/sim.1513) should be further emphasized. The abstract and discussion focus too heavily on the prevalence estimates without considering these limitations, emphasizing findings such as "more than half of young people" having significant symptoms of depression or anxiety, without assessing whether the subsample was representative or the symptoms were clinically meaningful with impact on functioning. The findings related to COVID-related and pandemic/infodemic factors associated with mental health are considerably more compelling than the potentially biased prevalence estimates.
  2. There appear to be some inaccuracies in the referencing. For example, "Further details concerning the goals, objectives 182 and methods of this broad research project can be found elsewhere [23-24]" - citation 24 seems unrelated to the current study. Please carefully recheck references.
  3. It is unclear why the analyses only investigated PHQ and GAD at the clinical cut points. No rationale is given for the focus on possible MDE/GAD rather than using the scores on the scales as indicative of symptom severity (which is likely to be more appropriate given the non-clinical nature of the study).
  4. Were prevalence estimates based on weighted or unweighted data?
  5. The references to "wave 1" and "wave 2" are a little unclear - referring instead to the time period (ie April-June 2020 vs November 2020) would help to contextualize the findings. Moreover, cross-national differences in virus incidence, lockdown rules and other public health responses are likely to have been substantial. While analyses adjust for overall prevalence etc across the period April-November, changes in public messaging and risk over time within this period were not assessed contemporaneously to the surveys.
  6. Tense use is inconsistent in the results section (should be past tense).
  7. Why are GAD and MDE collapsed into a single variable for the models in Table 3 and 5? Did the authors examine differences in patterns of anxiety and depression symptoms? What was the correlation between PHQ-9 and GAD-7 scores?
  8. There is quite a lot of redundancy in the analyses. The two waves are treated separately, rather than as an independent variable. Both partially- and fully-adjusted models are presented - Tables 3 and 4 seem somewhat redundant as these models don't account for confounding as adequately as the model in Table 5.
  9. Without baseline pre-COVID data on depression and anxiety (and unclear representativeness of the samples), it is difficult to say how much prevalence increased from April/June – November, and harder still to attribute any changes to COVID. The authors should be more cautious in interpreting changes in prevalence within repeated cross-sections in the abstract and throughout the discussion. Such differences could be due to sampling or methodological biases. Without a longitudinal sample, change can only be weakly inferred.
  10. Discussion: How were the "three most important factors" determined? Did the authors examine whether the magnitude of the effect sizes was significantly different from other factors? Could the magnitude of ORs be related to scaling effects (ie some items were yes/no whereas others had Likert scales that allows extremes to be examined). Despite these concerns, I appreciate that the authors focus on the effects with strongest relationships, given many of the effects were modest (despite being significant).
  11. The discussion indicates that GINI index was associated with GAD or MDE. This is not consistent with Table 5, which appears to show no significant association. Did the authors also consider accounting cultural factors such as individualism/collectivism at the national (or individual) level?

Round 2

Reviewer 2 Report

The authors have been highly responsive to my suggestions.

I understand why the authors have chosen to use cut-points for depression and anxiety, despite losing much of the variability seen in the continuous scores. A little more acknowledgement of the limitations of using the cut-points, particularly in population-based data, should be provided. There is an recent review of this topic that could be cited: https://doi.org/10.1016/j.jclinepi.2020.02.002 - indicating that prevalence estimates are likely to be approximately double what would seen when using clinical diagnosis as an outcome (see also https://doi.org/10.1503/cmaj.110829). Such limitations would not be an issue if the continuous scores were used for the analyses. 

Some additional acknowledgement of the limitations of repeated cross-sectional data would also be appropriate, given that longitudinal studies on this topic have been published, e.g., https://doi.org/10.1192/bjp.2020.212, https://doi.org/10.1016/S2215-0366(20)30482-X, https://doi.org/10.1016/j.bbi.2020.04.028, https://doi.org/10.1016/j.janxdis.2021.102377, http://doi.org/10.5694/mja2.51043. 
